# Development of an In-Field Real-Time LAMP Assay for Rapid Detection of Tomato Leaf Curl New Delhi Virus

**DOI:** 10.3390/plants12071487

**Published:** 2023-03-29

**Authors:** Andrea Giovanni Caruso, Arianna Ragona, Sofia Bertacca, Mauricio Alejandro Marin Montoya, Stefano Panno, Salvatore Davino

**Affiliations:** 1Department of Agricultural, Food and Forest Sciences (SAAF), University of Palermo, Viale delle Scienze, 90128 Palermo, Italy; 2Laboratory of Industrial Microbiology, Faculty of Sciences, National University of Colombia, Calle 59A N.° 63-20, Medellín 050034, Colombia

**Keywords:** real-time LAMP, ToLCNDV, *Begomovirus*, in-field detection, zucchini squash

## Abstract

Tomato leaf curl New Delhi virus (ToLCNDV) represents a threat to economically important horticultural crops. A real-time loop-mediated isothermal amplification (LAMP) assay for in-field ToLCNDV detection was developed, coupled to a rapid sample preparation method, and tested both in field and laboratory conditions on zucchini squash, tomato, and pepper samples. A set of six LAMP primers was designed for specific ToCLNDV detection, targeting a 218-nucleotide sequence within the AV1 gene. The sensitivity, specificity and accuracy of the real-time LAMP assay and comparison with canonical PCR were evaluated. The real-time LAMP assay developed was about one-thousand times more sensitive than the conventional PCR method, detecting a total of 4.41 × 10^2^ genome copies as minimum target; no cross-reactivity was detected with the other geminiviruses used as the outgroup. The rapid sample preparation method allows for a reliable detection with a low reaction delay (≈2–3 min) compared to canonical DNA extraction, providing results in less than 45 min. Lastly, an increase in ToLCNDV-positive sample detection was observed compared to PCR, in particular for asymptomatic plants (85% and 71.6%, respectively). The real-time LAMP assay developed is a rapid, simple, specific, and sensitive technique for ToLCNDV detection, and it can be adopted as a routine test, for both in-field and laboratory conditions.

## 1. Introduction

Tomato leaf curl New Delhi virus (ToLCNDV) is a bipartite begomovirus (family: *Geminiviridae*) [1] which infects economically important crops belonging to *Cucurbitaceae* and *Solanaceae* families [2]. This pathogen is particularly concerning for cucurbitaceous crops, with special regard to zucchini squash (*Cucurbita pepo*). As reported by the European and Mediterranean Plant Protection Organisation, the number of new hosts of ToLCNDV increases every year [3]. In Italy, zucchini squash is an important horticultural crop that accounts for more than 19,950 ha in greenhouse and open field, and a total production of about 6230 tonnes in 2022 [4], although, in recent years, this crop has been constantly threatened by various pathogens, including ToLCNDV. The first report of ToLCNDV was recorded in Asia on tomato plants (*Solanum lycopersicum* L.) [5] and successively on other horticultural crops, such as zucchini squash, eggplant (*Solanum melongena* L.), potato (*Solanum tuberosum* L.), and chilli pepper (*Capsicum annuum* L.) [6].

Afterwards, it was detected in other countries on the Indian subcontinent (Bangladesh and Pakistan), in Southeast Asia (Philippines, Indonesia, Sri Lanka, Thailand, and Taiwan), and the Middle East (Iran) [7]. In the Mediterranean basin, ToLCNDV was detected in Spain, Italy, Tunisia, Morocco, Algeria, Greece, and France [8,9,10,11,12,13,14]. Regarding Italy, in October 2015, it was detected for the first time in the horticultural area of Trapani province (Sicily), where severe symptoms on zucchini squash in the open field were observed, not previously reported by growers [9]; subsequently, it was also found in Sardinia [15], Lazio [16], Calabria, Campania, and Apulia regions [2,17]. To date, due to the several outbreaks in different countries of the Mediterranean basin, ToLCNDV is included in the European A2 list of pests recommended for regulation as quarantine pests [18], as reported in the European Commission Implementing Regulation (EU) 2019/2072 [19].

The ToLCNDV genome consists of two circular single-stranded DNA components (ssDNA), DNA-A and DNA-B, of 2.7 and 2.6 kb, respectively [5]. The DNA A virion-sense strand encodes the coat protein (CP, ORF AV1/V1) and ORF AV2/V2, which is implicated in virus movement, while the complementary-sense strand encodes the replication-associated protein (Rep, ORF AC1/C1), a transcriptional activator protein (TrAP, ORF AC2/C2), a replication enhancer protein (REn, ORF AC3/C3), and C4 protein (ORF AC4/C4). DNA B encodes a nuclear shuttle protein (NSP, ORF BV1) on the virion-sense and a movement protein (MP, ORF BC1) on the complementary-sense [1].

DNA-A produces virions and can replicate autonomously; it also encodes all the information for viral encapsidation and replication. DNA-B encodes for two proteins involved in the intracellular and intercellular virus mobility [20]. DNA-B is required for systemic infection, systemic movement, symptom expression, and nuclear localisation, but it cannot replicate in the absence of DNA-A [21]. These two genomic components present an intergenic region (IR) with motifs required to control the gene expression along with the initiation of replication. The common region (CR) contained in the intergenic region is present in both DNA components and shows a main topological feature: a hairpin structure with a conserved nonanucleotide sequence (TAATATT↓AC) that spans the origin of virion replication strand (ORI, indicated by ↓) [5].

Zucchini squash and tomato are the most economically important species in Italy affected by this disease. On zucchini squash plants, the characteristic symptoms caused by ToLCNDV infection are yellow mosaic, severe leaf curling, vein swelling in young leaves, and internode shortening, while on cucurbit fruits, skin roughness, longitudinal cracking, and reduced size have been observed. The disease is quite pronounced, especially in very young seedlings; in adult plants, already formed fruits show pronounced deformation and surface roughness [2]. On tomato, symptoms consist mainly of chlorotic mottling, dark greening, downward curling of the leaves, and vein distortion on middle and lower leaves [5,22].

ToLCNDV is transmitted by the whitefly *Bemisia tabaci* Gennadius (Hemiptera: Aleyrodidae), widely distributed in tropical and subtropical regions and in the Mediterranean basin [23], representing a major threat to different cultivated crops. In general, *B. tabaci* transmits the begomoviruses in a circulative persistent manner, which are mostly restricted to the phloem of infected plants [24]. Regarding the seed transmission, there are currently doubts about this transmission mode; although some authors have experimentally demonstrated the possibility of ToLCNDV transmission through seeds [7], it is expected to be of little relevance under field conditions. In contrast, mechanical inoculation successfully transmitted three different Italian ToLCNDV isolates to *C. pepo*, *C. melo inodorus*, and *C. melo cantalupensis* [2]. Moreover, López and co-workers [25] confirmed the mechanical transmission in 4 genera and 13 species belonging to the *Cucurbitaceae* family, including a crop-related exotic germplasm used for cucurbits breeding, while tolerance to mechanical transmission of ToLCNDV was identified in melon, within *Cucumis melo* subsp. *agrestis* var. *momordica* and in wild *agrestis* accessions.

Viruses are some of the most common and dangerous plant pathogens, causing significant economic losses worldwide [26], and it is extremely important to carry out appropriate crop management to avoid their spread. Virus disease management is essentially based on the application of preventive measures. In this context, limiting ToLCNDV infections is fundamental to control whiteflies, for example, using adult mass trapping, parasitoid/predator releases, insect-proof nets, and selective insecticides [27]; moreover, it is important to remove symptomatic plants to reduce the inoculum and consequently the virus spread to the whole crop.

Regarding the diagnosis, to date, serological and molecular methods are available for ToLCNDV detection. Serological methods include DAS-ELISA [28] and commercial lateral flow ImmunoStrip^®^ (LFA) (Agdia, Elkhart, IN, USA), while molecular methods include end-point PCR [2,22], rolling circle amplification PCR (RCA-PCR) [29], real-time PCR [30], nucleic acid spot hybridisation (NASH) using specific riboprobes [31], colorimetric loop-mediated isothermal amplification (LAMP) [32], LAMP-coupled CRISPR–Cas12a [33], and a commercial ready-to-use real-time LAMP kit combined with proprietary devices (i.e., an ICGENE ToLCNDV diagnostic kit).

In recent years, a real-time pathogen detection based on the LAMP technique made it possible to achieve a rapid and accurate diagnosis both in laboratories and in the field, representing a methodology to prevent the dispersion of endemic diseases and the introduction of dangerous pathogens [34,35] into new geographical areas [36]. Regarding the ToLCNDV detection, as described above, different LAMP assays have already been developed which, however, while maintaining a good degree of sensitivity and specificity compared with other diagnostic methods, may have some limitations in terms of applicability and easiness. In detail, colorimetric LAMP allows to obtain the results at the end of the test only through visual inspection by the operator using colorimetric indicators, while real-time assays, such as LAMP-coupled CRISPR–Cas12a, require more time for diagnosis, trained operators, and lower applicability in the field. Lastly, ready-to-use LAMP kits using commercial devices, although they can be easily used in field conditions, turn out to be closed systems, as they force the operator to use specific devices and kits for diagnosis.

In the present study, a rapid and sensitive protocol based on real-time LAMP assay for directly use for in-field ToLCNDV detection was developed; additionally, a simple and inexpensive sample preparation method was used.

## 2. Materials and Methods

### 2.1. Source of Viral Material

A total of three characterised ToLCNDV isolates were used to develop the real-time LAMP assay. In detail, total DNA was extracted from three lyophilised ToLCNDV isolates named Trapani1703 (Acc. No. MH475378), Napoli166 (Acc. No. MH475422), and PAV200 (Acc. No. KX826050) [2], stored at the SAAF Department (University of Palermo), using the GenUP^®^ Plant DNA kit (Biotechrabbit GmbH, Berlin, Germany) following the manufacturer’s instructions, with minor modifications. About 100 mg of lyophilised leaf tissue was homogenised in a sample extraction bag using the HOMEX 6 homogeniser (Bioreba, Reinach, Switzerland), with 1 mL extraction buffer (1.3 g sodium sulphite anhydrous, 20 g polyvinylpyrrolidone MW 24–40.000, 2 g chicken egg chicken albumin grade II, 20 g Tween-20 in one L of distilled water; pH 7.4). Afterward, aliquots of 400 µL of the sample extract were added to the same lysis buffer volume provided in the kit, and the manufacturer’s protocol was followed from this step. Lastly, each sample was resuspended in 100 µL RNase-free water. Total DNA from healthy zucchini squash leaves was also extracted and used as a negative control (NC). Total DNA concentration was determined in duplicate with a NanoDrop 1000 spectrophotometer (Thermo Fisher Scientific, Waltham, MA, USA). Dilutions were adjusted to ≈50 ng/µL and stored at −20 °C until subsequent analyses.

### 2.2. AV1 Gene Amplification by PCR 

Polymerase chain reaction assay was carried out on DNA of the three isolates previously mentioned. In order to amplify the AV1 (coat protein) gene, the primer pair ToLCNDV-CP1/ToLCNDV-CP2 [2] was used. PCR was performed in a final reaction volume of 25 µL, containing 2 µL of total DNA, 20 mM Tris-HCl (pH 8.4), 50 mM KCl, 3 mM MgCl_2_, 0.4 mM dNTPs, 1 µM of each primer, 2 U Taq DNA polymerase (Thermo Fisher Scientific, Waltham, MA, USA), and RNase-free water to reach the final volume. PCR was carried out in a MultiGene OptiMax thermal cycler (Labnet International Inc., Edison, NJ, USA) according to the following conditions: initial denaturation at 94 °C for 3 min; 40 cycles at 94 °C for 50 s, 52 °C for 1 min, and 72 °C for 2 min; and a final elongation at 72 °C for 10 min [2]. Molecular-grade water and total DNA extracted from healthy zucchini squash plant were used as negative controls. The PCR products were electrophoresed on 1.5% (*w*/*v*) agarose gel, stained with SYBR Safe (Thermo Fisher Scientific, Waltham, MA, USA), and visualised by UV light. Subsequently, the expected amplicons of 1050 bp were purified from agarose gel using an UltraClean^TM^ 15 DNA purification kit (MO-BIO Laboratories, Carlsbad, CA, USA), following the manufacturer’s instructions. The purified products were quantified twice using a UV–VIS NanoDrop 1000 spectrophotometer (Thermo Fisher Scientific, Waltham, MA, USA) and sequenced in both directions using an ABI PRISM 3100 DNA sequence analyser (Applied Biosystems, CA, USA).

### 2.3. Primer Design for LAMP Assay Set-Up

In order to design a LAMP primer set, the AV1 sequence of ToLCNDV Trapani1703 was used for the PrimerExplorer version 5 software (http://primerexplorer.jp/lampv5e/, accessed on 10 December 2022) as a reference sequence. A specific 450 bp nucleotide inside the AV1 gene was chosen as the amplification target. The primer set selected included a total of six primers: two external primers (forward outer primer F3 and backward outer primer B3), two internal primers (forward inner primer FIP and backward inner primer BIP), and two additional loop primers (forward loop primer LF and backward loop primer LB); the presence of loop primers significantly improve the efficiency, sensitivity, and selectivity of the reaction, reducing the time it takes by 50% [37]. In order to evaluate possible cross-reactions with other organisms, the specificity of each primer obtained was verified in silico using the Nucleotide-BLAST algorithm (BLASTn tool) (https://www.ncbi.nlm.nih.gov, accessed on 10 December 2022) available at the National Centre for Biotechnology Information (NCBI). Moreover, to verify the primer set affinity, each primer was tested using the Vector NTI Advance 11.5 software (Invitrogen, Carlsbad, CA, USA) against the full genomic sequences of the following geminiviruses: Tomato yellow leaf curl virus (TYLCV) (GenBank Acc. No. X15656), TYLCV-IL23 (GenBank Acc. No. MF405078), TYLCV isolate 8–4/2004 (GenBank Acc. No. DQ144621), Tomato yellow leaf curl Sardinia virus (TYLCSV) (GenBank Acc. No. GU951759), Potato yellow mosaic Panama virus (DNA-A GenBank Acc. No. NC_002048, DNA-B GenBank Acc. No. NC_002049), Tomato leaf curl Sinaloa virus (DNA-A GenBank Acc. No. NC_009606, DNA-B GenBank Acc. No. NC_009605), and Tomato yellow mottle virus (DNA-A GenBank Acc. No. KC176780, DNA-B GenBank Acc. No. KC176781).

### 2.4. ToLCNDV Real-Time LAMP Assay Optimisation

Real-time LAMP reactions were performed on a Rotor-Gene Q 2plex HRM Platform Thermal Cycler (Qiagen, Hilden, Germany) using LAMP Isothermal Master Mix from OptiGene (West Sussex, United Kingdom). Each isothermal reaction was performed in a final volume of 25 μL, containing 0.2 μM each of ToLCNDV-F3 and ToLCNDV-B3, 1.6 μM each of ToLCNDV-FIP and ToLCNDV-BIP, 0.4 μM each of ToLCNDV-LF and ToLCNDV-LB, 15 μL of LAMP Isothermal Master Mix (Optigene Limited, West Sussex, United Kingdom), 2 μL of total DNA (50 ng/μL) as template, and nuclease-free water to reach the final volume. The three isolate samples, previously described and verified by end-point PCR, were initially used for the real-time LAMP assay. The real-time LAMP assay was conducted at 65 °C (according to manufacturer’s instructions) for 60 min, and fluorescence was acquired every 60 s; additional steps for melting curve were carried out to record the fluorescence (95 °C for 1 min, 40 °C for 1 min, 70 °C for 1 min and an increase in temperature at 0.5 °C/s up to 95 °C). The relative fluorescence unit (RFU) threshold value was used, and the threshold time (Tt) was calculated as the time at which fluorescence was equal to the threshold value, while the fluorescence data were obtained in the 6-carboxyfluorescein (FAM) channel according to the manufactures’ instructions (excitation at 450–495 nm and detection at 510–527 nm) during the amplification. Each sample was analysed in triplicate in three independent real-time LAMP assays. In each run, total DNA from healthy zucchini squash plant was included as a negative control (NC).

### 2.5. Sensitivity, Reaction Time, and Specificity of Real-Time LAMP Assay and Comparison to End-Point PCR

The purified PCR products previously obtained (see Section 2.2) were used to set up the real-time LAMP assay conditions. The number of copies of each sample was determined by the following formula: number of copies = (amount of DNA in nanograms × 6.022) × 10^23^)/(length of DNA template in bp × 1 × 10^9^ × 650). Tenfold serial dilutions of each sample diluted into healthy zucchini squash plant extracts were used as a template to ascertain the optimal reaction time and sensitivity of the real-time LAMP for ToLCNDV detection. In addition, real-time LAMP and end-point PCR (see Section 2.2) assays were carried out to compare their sensitivity. The real-time LAMP assay specificity and the potential non-specific reactions with other viruses were determined; in this case, a real-time LAMP test was conducted including three ToLCNDV-positive samples, DNA of other geminiviruses used as outgroup (Tomato yellow leaf curl virus (TYLCV) (GenBank Acc. No. X15656), TYLCV-IL23 (GenBank Acc. No. MF405078), TYLCV isolate 8-4/2004 (GenBank Acc. No. DQ144621), Tomato yellow leaf curl Sardinia virus (TYLCSV) (GenBank Acc. No. GU951759), Potato yellow mosaic Panama virus (DNA-A GenBank Acc. No. NC_002048, DNA-B GenBank Acc. No. NC_002049), Tomato leaf curl Sinaloa virus (DNA-A GenBank Acc. No. NC_009606, DNA-B GenBank Acc. No. NC_009605), Tomato yellow mottle virus (DNA-A GenBank Acc. No. KC176780, DNA-B GenBank Acc. No. KC176781), and total DNA from a healthy zucchini squash plant (NC). Each sample was analysed in triplicate in three independent real-time LAMP assays, and the assay was conducted as above described (see Section 2.4), including the melting curve steps.

### 2.6. Rapid Sample Preparation Method Suitable for the ToLCNDV Real-Time LAMP Assay

A simple and inexpensive sample preparation procedure named “membrane spot crude extract” [38] that permits the setting up of a rapid LAMP assay and avoiding the canonical DNA extraction was used. In detail, ≈100 mg of lyophilised vegetable tissue of each isolate previously analysed by end-point PCR and real-time LAMP were homogenised in a sample extraction bag with 3 mL of extraction buffer. A total of 5 µL of extract was spotted on a 1 cm^2^ Hybond^®^-N+ hybridisation membrane (GE Healthcare, Chicago, IL, USA), dried at room temperature for 5 min, and placed in a 2 mL tube containing 250 µL of glycine buffer (0.1 M glycine, 0.05 M NaCl, 1 mM EDTA). After 20 s vortexing, samples were heated at 95 °C for 10 min, and 3 μL of the extract was used for the real-time LAMP assay, in a final reaction volume of 25 µL, including a healthy zucchini squash plant DNA as a negative control (NC). Each sample was analysed in triplicate in three independent real-time LAMP assays, comparing the results of the “membrane spot crude extract” method with the total DNA extraction performed with commercial kits (see Section 2.1).

### 2.7. Direct on Field Analysis of Tomato Leaf Curl New Delhi Virus

In order to assess the in-field application of the real-time LAMP analysis developed in this work, the ready-to-use real-time LAMP reactions, disposables, and tools listed in Table 1 were prepared.

In detail, 20 zucchini squash plants (15 symptomatic and 5 asymptomatic) in an open field cultivation were investigated. From each plant, a total of 7 leaf disks of ≈0.5 cm^2^ of diameter obtained using a single hole puncher were collected (disinfecting the tool between each sample), placed in a sample extraction bag (BIOREBA, Reinach, Switzerland) with 2 mL extraction buffer (1.3 g sodium sulphite anhydrous, 20 g polyvinylpyrrolidone MW 24–40,000, 2 g chicken egg chicken albumin grade II, 20 g Tween-20 in one L of distilled water; pH 7.4), and homogenised using a hand homogeniser (BIOREBA, Reinach, Switzerland). A total of 5 µL of the obtained homogenate was spotted in ≈0.5 cm^2^ of Hybond^®^-N+ hybridisation membrane (GE Healthcare, Chicago, IL, USA) directly placed inside a 2 mL tube and dried at room temperature for 5 min. Subsequently, 250 µL of glycine buffer (0.1 M glycine, 0.05 M NaCl, 1 mM EDTA) was added. After 20 s of mixing by inversion, 3 µL was used for the real-time LAMP reaction, performed in a bCube2 system (Hyris Ltd., London, United Kingdom). In this case, the bCube2 instrument was supplied with a 12 Volt (12 Ah) rechargeable battery. The same experiment just described was also performed in a different open field, considering a total of 20 tomato plants and 20 pepper plants (15 symptomatic and 5 asymptomatic each). The sampling areas selected were marked by GPS using the Planthology mobile application [39]. Simultaneously, plant materials from the same plants were also collected and analysed in laboratory conditions by canonical end-point PCR (using total DNA) and real-time LAMP assays (using both total DNA and membrane spot crude extract), using a Rotor-Gene Q 2plex HRM Platform thermal cycler (Qiagen, Hilden, Germany). A positive control (PC) and healthy zucchini squash/tomato/pepper plant DNA as negative controls (NC) were included in each test. All the protocols used are reported in Section 2.2 and Section 2.4.

## 3. Results

### 3.1. AV1 Gene Amplification by PCR

The obtained PCR products from the three isolates named Trapani1703, Napoli166, and PAV200 used in this study showed the expected 1050 bp amplicon size. The obtained sequence showed a percentage identity of >99.9% with the previously ToLCNDV sequences deposited in GenBank [2].

### 3.2. Primers Design for LAMP Assay Set-Up

A real-time LAMP assay for the rapid ToLCNDV detection was developed using a set of six primers designed on the ToLCNDV-AV1 coding region. The primer sequences and binding sites are shown in Table 2 and Figure 1, respectively.

The in silico analysis carried out using the Nucleotide-BLAST algorithm (BLASTn tool) showed no cross-reactions with other organisms, while the hybridisation analysis performed with the Vector NTI 11.5 program excluded relevant matches with the other geminiviruses reported in Section 2.3.

### 3.3. ToLCNDV Real-Time LAMP Assay Optimisation

The real-time LAMP assay was conducted using the DNA extracted from the three samples previously analysed by end-point PCR, including a healthy zucchini squash plant DNA as a negative control (NC), to evaluate the performance of the primer set designed for ToLCNDV detection. As reported in Table 3 and Figure 2A, the amplification curves of positive samples reached the plateau in about 15 min. The melting curves of the positive real-time LAMP reactions displayed the same peak temperature of approximately 84.5 °C (Figure 2B). According to the end-point PCR results, no signal was obtained with the negative control (NC), even at late reaction times, as expected.

### 3.4. Sensitivity, Reaction Time, and Specificity of Real-Time LAMP Assay and Comparison to End-Point PCR

Tenfold serial dilutions of amplicons obtained by end-point PCR from the three ToLCNDV isolates were used as a template, starting from a concentration of 50 ng/μL each (4.41 × 10^10^ copies), to conduct a comparative experiment between the real-time LAMP and the end-point PCR assay sensitivity, and to evaluate the real-time LAMP efficacy. As reported in Table 4 and Figure 3, the real-time LAMP assay was able to detect DNA concentration up to ≈50 × 10^-8^ ng/μL, while DNA concentration up to ≈50 × 10^-5^ ng/μL was detected with the end-point PCR. The results of LAMP reaction time plateau, shown in Table 4, were calculated as the mean values obtained from the three replicates.

In addition, the results clearly showed that the time required to carry out the experiment was less than 45 min (Figure 3), even considering the lowest detectable concentration of the samples in real-time LAMP (≈50 × 10^-8^ ng/μL). Furthermore, to evaluate the specificity of the real-time LAMP assay and to assess potential nonspecific cross-reactions with other viruses, a real-time LAMP assay using the outgroup reported in Section 2.5 was performed. In this case, no signals were obtained with any of the geminiviruses used as outgroups, while a single peak at ≈84.5 °C in the melting curve only for the three ToLCNDV isolate samples was obtained, confirming the assay specificity. The LAMP assay was able to detect a total of 4.41 × 10^2^ genome copies, whereas with the canonical PCR, it was only possible to detect up to 4.41 × 10^5^ genome copies, demonstrating that the real-time LAMP assay developed in this work is significantly more sensitive.

### 3.5. Rapid Sample Preparation Method Suitable for the ToLCNDV Real-Time LAMP Assay

With the purpose of speeding up the real-time LAMP assay developed in this work and making it totally independent from canonical DNA extraction, a method that allowed for a simple and inexpensive sample was successfully applied. Therefore, the three ToLCNDV isolates (Table 2), previously analysed by both molecular assays, were considered. As reported in Table 5, the rapid method allows for the detection of ToLCNDV presence with a delay of only few minutes (≈2–3 min) compared to the corresponding total DNA obtained with the commercial kit (Figure 4). The results of the two extraction methods, shown in Table 5, were calculated as the mean values obtained from the three replicates. 

In detail, the three isolates resulted as positive in the LAMP assay when extracted with either procedure. Specifically, isolates extracted with the commercial kit and the “membrane spot crude extract” method showed a fluorescence increase ranging from 4 to 5.1 min and from 6.1 to 8.6 min, respectively, while the reaction plateau was reached in about 10 minutes and about 13–19 min, respectively. Lastly, no amplification was obtained with the negative control, even with this rapid procedure, as expected.

### 3.6. Direct In-Field Analysis of Tomato Leaf Curl New Delhi Virus

The in-field application of the real-time LAMP analysis developed in this work, using a rapid sample preparation method, was proven to be suitable for specific and sensitive ToLCNDV detection (Figure 5), providing results in less than 45 min (Table 6).

In detail, the real-time LAMP assay was proven to be reliable both in the field and in laboratory conditions, with fully comparable results. A total of 43 out of 60 samples resulted in being positive to ToLCNDV by end-point PCR, while 51 out of 60 samples provided positive results to ToLCNDV infection by real-time LAMP assay. The real-time LAMP assay showed an increase in detecting ToLCNDV-positive samples, in particular for asymptomatic plants; the percentage of samples that resulted positive were 85% and 71.6% with real-time LAMP and conventional RT-PCR assays, respectively. In detail, a total of 17 out of 20 zucchini squash samples resulted positive for ToLCNDV both in-field and laboratory conditions by real-time LAMP assay. In this case, unlike PCR, the LAMP assay was able to detect ToLCNDV infection in two zucchini squash asymptomatic plants. Regarding tomato and pepper samples, a total of 16 out of 20 and 14 out of 20 samples resulted positive for ToLCNDV infection, respectively; in these cases, the LAMP assay was able to detect the infection in a total of four tomato and two pepper asymptomatic plants. Lastly, to avoid false positive results, the melting curve confirmed the results, showing the same peak temperature (≈84.5 °C) in all analysed samples, including those that were asymptomatic and resulted positive only by LAMP, concordant with the ToLCNDV-positive controls.

## 4. Discussion

Every year, plant viruses affect several crops, causing considerable losses in terms of production and economic profit. The complexity of these pathogens makes disease management particularly difficult due to their dynamic epidemiology, driven by new emergence, outbreaks, and adaptability [40,41]. In particular, horticultural crops are constantly threatened by different viral pathogens [42]; in the last years, the spread of new viruses and the re-emergence of the ones already present in the Mediterranean basin emphasise the need for their adequate control [43,44]. Among these, Tomato leaf curl New Delhi virus represents an important threat to horticultural production, such as zucchini squash, tomato, and pepper; its host range and spread to new geographical regions are rapidly extending, especially in the Middle East and the western Mediterranean Basin, as well as for the wide spread of its insect vector *B. tabaci*, which remains viruliferous throughout its life span after feeding on the phloem sap of infected plants [45]. In order to avoid pesticide overuse and properly control the insect vector, experimental results showed that the best strategy for virus suppression is the Integrated Pest Management (IPM), regulated in the European Union Member States as an interdisciplinary approach for agricultural pest population control (Directive 2009/128/EC) [46], in which early diagnosis of plant material plays a major role.

In this context, disease management must be based on specific and sensitive diagnostic and detection tools in order to control the viral diseases dispersion. Throughout the years, plant virus detection was mainly based on enzyme-linked immunosorbent assay (ELISA) [47], which permits the analysis of numerous samples simultaneously, and molecular methods, such as molecular hybridisation [48] and PCR-based methods [49], that allow further levels of specificity and sensitivity compared to ELISA test, but are conversely more prone to false positives by contamination or false negatives by inhibitors and are less appropriate for massive sample analyses [48]. Regarding molecular diagnostic methods, the LAMP technique has proven to be sensitive, specific, and easy to use for the detection of several pathogens, showing optimal characteristics, even in field conditions, and reduced sensitivity to inhibitors [36], such as phenols and polysaccharides [50], compared to PCR-based methods, which requires nucleic acid extraction methods from plant samples [51].

In this work, we selected the ToLCNDV-AV1 coding region as the target for primer LAMP design for the absence of recombination events and a very low differentiation in several ToLCNDV isolates on the AV1 gene, as reported by Panno and co-workers [2]. The set of the six primers obtained showed good efficiency, sensitivity, and selectivity for ToLCNDV detection.

LAMP assay specificity is crucial to achieve a correct ToLCNDV discrimination from other viruses and has a high sensitivity to minimise occurrences of false negatives. The results of this work showed that the LAMP assay was about one-thousand times more sensitive than the conventional PCR method. The assay optimisation revealed that the analysis can be carried out in 45 min to obtain reliable results, greatly simplifying the in-field diagnosis. In addition, the LAMP assay could detect the virus presence from infected samples in as little as 10–15 min. The quick detection procedure developed for direct in-field diagnosis proved to be particularly reliable and simple. Through the use of a portable battery-powered thermal cycler, the analysis can be carried out without the need of a power grid; moreover, the simplicity of sample preparation and a ready-to-use LAMP reaction mixture did not require the presence of skilled personnel or specific laboratory equipment.

Moreover, the data obtained from the comparison of conventional extraction using a commercial kit and the “membrane spot crude extract” method suggest that the LAMP-based detection method could be suitable for direct use in the field, thanks to the rapid sample preparation method and the drastic reducing of the analysis cost. For that reason, the real-time LAMP technique developed in this work, associated with the rapid sample extraction method, could be an efficient tool for a rapid on-site detection, simplifying the surveys of the ToLCNDV spread. The effectiveness of the technique developed is essential to understand the pathogen spread and to prevent the quickly in field ToLCNDV dispersion.

## 5. Conclusions

Important improvement has been made throughout the years in control strategies to reduce plant disease damage. In particular, viral diseases can be effectively controlled through the development of new diagnostic protocols as a useful step towards the containment of new epidemics. The real-time LAMP assay developed in this work allows for the processing of a great number of samples at the same time directly in the field, especially when associated with a rapid sample preparation method, resulting in a rapid, simple, specific, and sensitive technique for detecting the ToLCNDV presence. Therefore, it could be interesting to adopt this method as a routine test for a suitable ToLCNDV detection, both under laboratory and field conditions, representing a reliable tool for large virus surveys.

## Figures and Tables

**Figure 1 plants-12-01487-f001:**
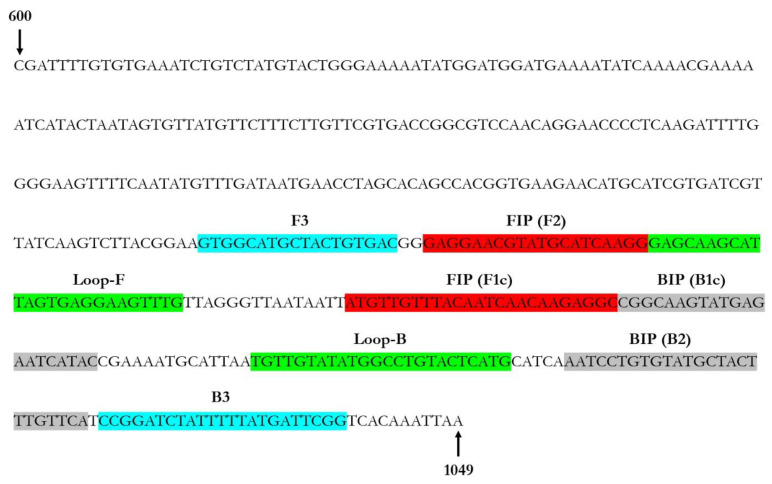
Loop-mediated isothermal amplification (LAMP) primer set location designed on the ToLCNDV-AV1 coding region. F3 and B3 are shown in pale blue, FIP (F1c-F2) in red, BIP (B1c-B2) in grey, and the two loop primers LF and LB in green. FIP is a hybrid primer consisting of the F1c and the F2 sequences, while BIP is a hybrid primer consisting of the B1c and B2 sequences. The numbers at the beginning and end of the sequence represent the genomic position of the first and last nucleotide in the selected ToLCNDV DNA-A complete sequence (GenBank Acc. No. MK732932).

**Figure 2 plants-12-01487-f002:**
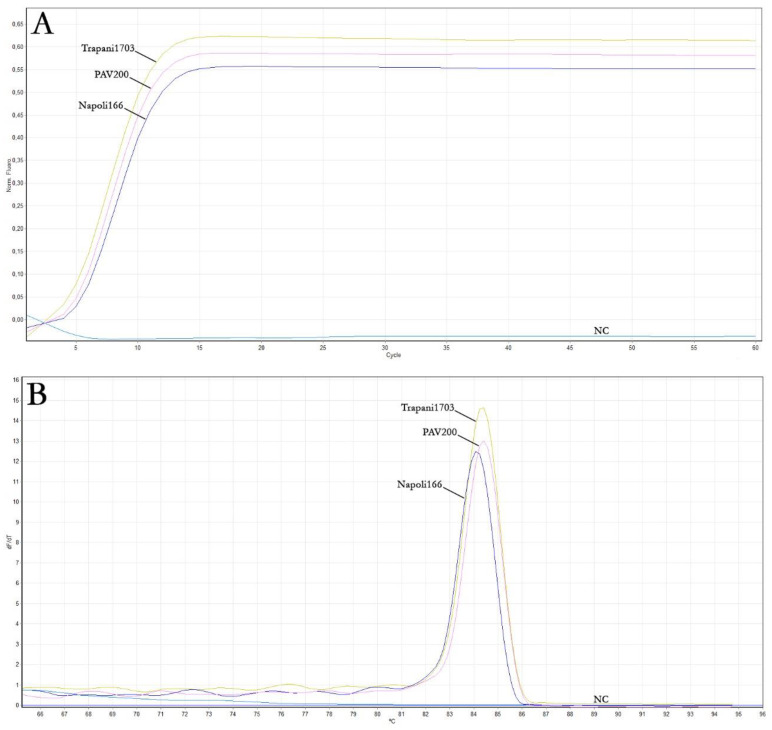
Results of the real-time LAMP assay for the detection of ToLCNDV. (**A**) Real-time LAMP amplification curves of the three ToLCNDV isolates. (**B**) Melting curves. NC: negative control.

**Figure 3 plants-12-01487-f003:**
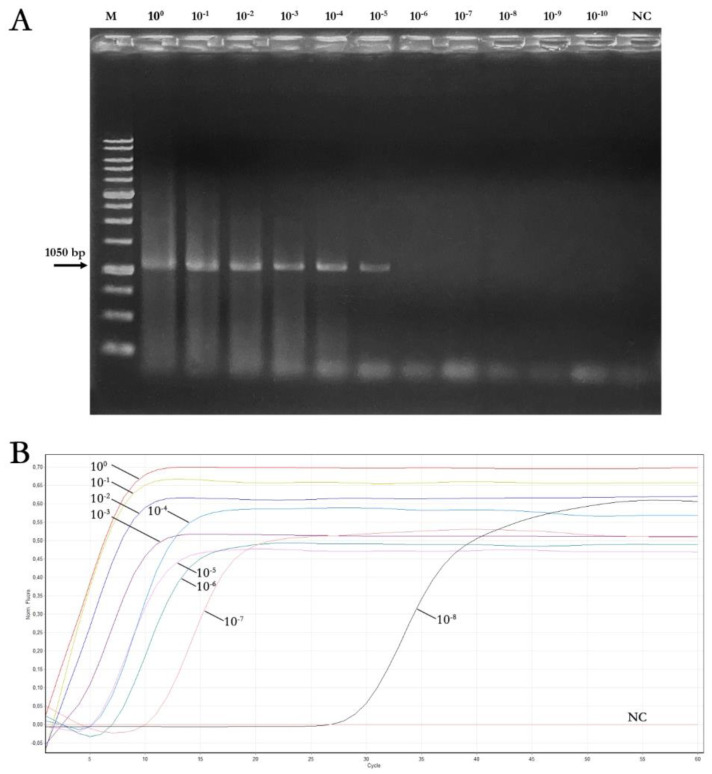
Sensitivity of the end-point PCR and real-time LAMP assays for ToLCNDV detection using 10-fold serial dilution PCR products of three ToLCNDV isolates (Trapani1703 isolate showed in this figure). (**A**) Agarose gel electrophoresis of PCR products. M: Nippon Genetics 1 Kb ladder RTU, NC: negative control. (**B**) Fluorescence of the 10-fold serial dilutions analysed. Fluorescence increased in positive sample curves (from ≈50 × 10^-1^ ng/μL to ≈50 × 10^-8^ ng/μL) after 2 to 30 min.

**Figure 4 plants-12-01487-f004:**
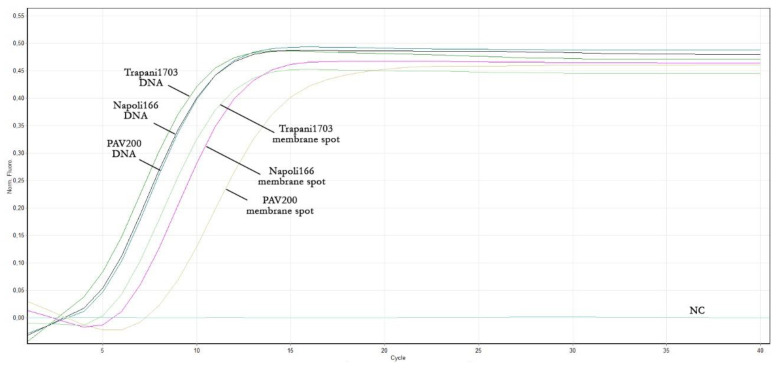
Real-time LAMP amplification curves of the three ToLCNDV isolates starting from total DNA obtained with a commercial kit and the membrane spot crude extract method.

**Figure 5 plants-12-01487-f005:**
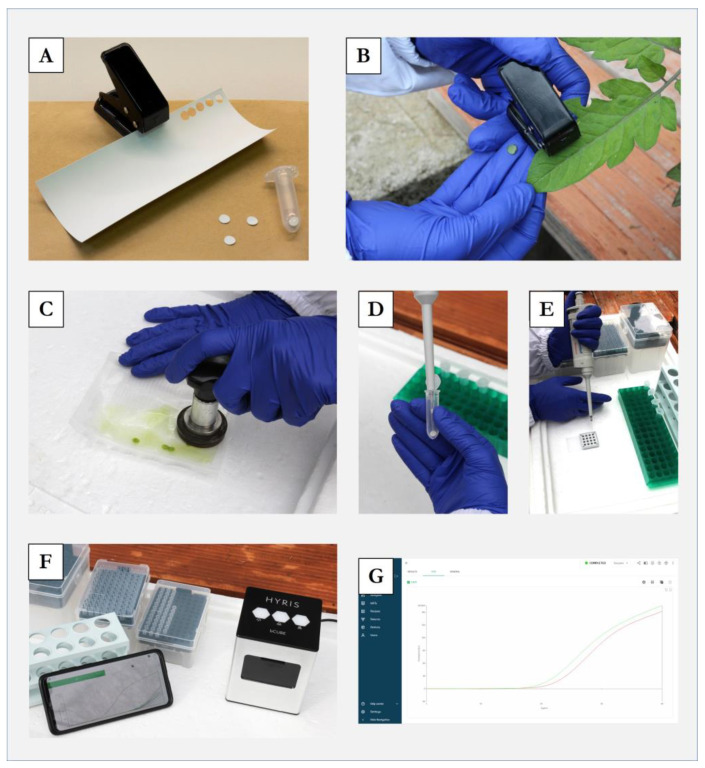
In-field detection of tomato leaf curl New Delhi virus (ToLCNDV) using a real-time LAMP assay and a rapid sample preparation method. (**A**) Previous preparation of ≈0.2 cm^2^ of a hybridisation membrane directly placed inside a sterile 2 mL tube. (**B**) Leaf disk sampling using a single hole puncher. (**C**) Sample homogenisation using a hand homogeniser. (**D**) Homogenate spot in hybridisation membrane–glycine buffer addition. (**E**) Sample loading. (**F**) Real-time LAMP analysis using a portable battery-powered thermal cycler. (**G**) Result visualisation in real time via mobile devices.

**Table 1 plants-12-01487-t001:** List of reagents, disposables, and tools used for in-field ToLCNDV detection by real-time LAMP assay.

Item	Quantity
Sample extraction bag with 2 mL extraction buffer	20
Single hole puncher	1
2 mL tubes with ≈0.2 cm^2^ hybridisation membrane	20
Hand homogeniser	1
Glycine buffer	6 mL
LAMP reaction mixture	550 µL
Positive control (PC)	1
Negative control (NC)	1
bCube2 thermal cycler	1
16-well bCube cartridge	1
bCube cartridge aluminium foil	1
12 Volt rechargeable battery	1
Smartphone/laptop	1
100–1000 µL micropipette	1
10–100 µL micropipette	1
0.5–10 µL micropipette	1
100–1000 µL micropipette tips	1 rack (96 tips)
10–100 µL micropipette tips	1 rack (96 tips)
0.5–10 µL micropipette tips	1 rack (96 tips)

**Table 2 plants-12-01487-t002:** LAMP primers for ToLCNDV detection designed in this study.

Primer Name	Length (nt)	Sequence 5′-3′	Nucleotide Position (nt)	Amplicon Size (bp)	Reference Sequence (Acc. No.)
ToLCNDV-F3	18	GTGGCATGCTACTGTGAC	821–838	218	MK732932
ToLCNDV-B3	23	CCGAATCATAAAAATAGATCCGG	1016–1038		
ToLCNDV-BIP	46	CGGCAAGTATGAGAATCATACTGAACAAAGTAGCATACACAGGATT	926–971	–	
ToLCNDV-FIP	45	GCCTCTTGTTGATTGTAAACAACATGAGGAACGTATGCATCAAGG	881–925		
ToLCNDV-LF	25	CAAACTTCCTAACTAATGCTTGCTC	861–885	–	
ToLCNDV-LB	24	TGTTGTATATGGCCTGTACTCATG	961–984		

**Table 3 plants-12-01487-t003:** Real-time LAMP assay performance for ToLCNDV detection.

Sample	Real-Time LAMP Reaction Time (min)		
	Assay #1	Assay #2	Assay #3
ToLCNDV Trapani1703	14	14.8	14.2
ToLCNDV Napoli166	16	15.4	14.9
ToLCNDV PAV200	15	15.8	15.3
Negative control (NC)	-	-	-

**Table 4 plants-12-01487-t004:** Comparison of real-time LAMP and end-point PCR sensitivity.

	Starting DNA Concentration (50 ng/μL) (4.41 × 10^10^ Copies)										
Assay	10^0^	10^−1^	10^−2^	10^−3^	10^−4^	10^−5^	10^−6^	10^−7^	10^−8^	10^−9^	10^−10^
End-point PCR	+	+	+	+	+	+	−	−	−	−	−
LAMP reaction time plateau (min) (mean values ± SD)	10.3 ± 0.2	10.3 ± 0.2	11 ± 0.2	14 ± 0.3	16.2 ± 0.3	15.9 ± 0.5	19.1 ± 0.2	24.9 ± 0.1	45 ± 0.2	−	−

**Table 5 plants-12-01487-t005:** Real-time LAMP assay performance comparison of different sample extraction methods for ToLCNDV detection.

Sample	Reaction Time Plateau (min) (Mean Values ± SD)	
	Total DNA (Commercial Kit)	Membrane Spot Crude Extract
ToLCNDV Trapani1703	13.1 ± 0.3	15.2 ± 0.3
ToLCNDV Napoli166	13.2 ± 0.3	16.2 ± 0.5
ToLCNDV PAV200	12.9 ± 0.2	20.1 ±0.3
Negative control (NC)	-	-

**Table 6 plants-12-01487-t006:** Results of in-field real time LAMP detection of ToLCNDV using the rapid sample preparation method of symptomatic and asymptomatic field samples (zucchini squash, tomato, and pepper) and comparison to end-point PCR and real-time LAMP (using both total DNA and membrane spot crude extract) in laboratory conditions. Asymptomatic samples detected by real-time LAMP assay are underlined in grey.

Sample Source	ID Sample	Symptomatic	In-Field Conditions	Laboratory Conditions	
			Real-Time LAMP Results	End Point PCR Results	Real-Time LAMP Results (Total DNA/Membrane Spot Crude Extract)
Zucchini squash	ZS-01	+	+	+	+/+
	ZS-02	−	+	−	+/+
	ZS-03	−	+	−	+/+
	ZS-04	+	+	+	+/+
	ZS-05	+	+	+	+/+
	ZS-06	+	+	+	+/+
	ZS-07	−	−	−	−
	ZS-08	+	+	+	+/+
	ZS-09	−	−	−	−
	ZS-10	+	+	+	+/+
	ZS-11	+	+	+	+/+
	ZS-12	+	+	+	+/+
	ZS-13	+	+	+	+/+
	ZS-14	+	+	+	+/+
	ZS-15	−	−	−	−
	ZS-16	+	+	+	+/+
	ZS-17	+	+	+	+/+
	ZS-18	+	+	+	+/+
	ZS-19	+	+	+	+/+
	ZS-20	+	+	+	+/+
	ZS-NC	−	−	−	−
Tomato	Tom-01	−	+	−	+/+
	Tom-02	−	−	−	−
	Tom-03	+	+	+	+/+
	Tom-04	−	−	−	−
	Tom-05	+	+	+	+/+
	Tom-06	+	+	+	+/+
	Tom-07	−	+	−	+/+
	Tom-08	+	+	+	+/+
	Tom-09	+	+	+	+/+
	Tom-10	−	+	−	+/+
	Tom-11	+	+	+	+/+
	Tom-12	+	+	+	+/+
	Tom-13	+	+	+	+/+
	Tom-14	+	+	+	+/+
	Tom-15	+	+	+	+/+
	Tom-16	−	−	−	−
	Tom-17	−	−	−	−
	Tom-18	−	+	−	+/+
	Tom-19	+	+	+	+/+
	Tom-20	+	+	+	+/+
	Tom-NC	−	−	−	−
Pepper	Pep-01	+	+	+	+/+
	Pep-02	+	+	+	+/+
	Pep-03	+	+	+	+/+
	Pep-04	+	+	+	+/+
	Pep-05	−	+	−	+/+
	Pep-06	+	+	+	+/+
	Pep-07	+	+	+	+/+
	Pep-08	+	+	+	+/+
	Pep-09	−	−	−	−
	Pep-10	−	−	−	−
	Pep-11	−	−	−	−
	Pep-12	+	+	+	+/+
	Pep-13	+	+	+	+/+
	Pep-14	−	+	+	+/+
	Pep-15	−	−	−	−
	Pep-16	+	+	+	+/+
	Pep-17	−	−	−	−
	Pep-18	+	+	+	+/+
	Pep-19	+	+	+	+/+
	Pep-20	−	−	−	−
	Pep-NC	−	−	−	−
ToLCNDV-positive control	PC	/	+	+	+/+

## Data Availability

Not applicable.

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
