# Peer review of "Development of an In-Field Real-Time LAMP Assay for Rapid Detection of Tomato Leaf Curl New Delhi Virus"

_plants, 2023, doi:10.3390/plants12071487_

Round 1

Reviewer 1 Report

In the present study, authors have developed a LAMP-based assay to study the plants infected by the leaf curl New Delhi virus. They have also shown the specificity and sensitivity of the method for detection of the ToLCNDV in the laboratory as well as in field conditions. There are a few minor comments I want the authors to address:

1.       Since AV1 (coat protein) gene was used as a target for assay related information about this gene has to be mentioned in the introduction. additionally in the discussion mention why this particular gene has been chosen.

2.       For real-time LAMP assay the authors have used two replications rather than three. It will be good to add the results of another replication to enhance the robustness of the study.

3.       Lines from 363-366 “specifically……respectively”. Rephrasing is required for clarity better to write the fluorescence rise and plateau of commercial and membrane spots separately for more clarity.

Author Response

In the present study, authors have developed a LAMP-based assay to study the plants infected by the leaf curl New Delhi virus. They have also shown the specificity and sensitivity of the method for detection of the ToLCNDV in the laboratory as well as in field conditions. There are a few minor comments I want the authors to address:

  1. Since AV1 (coat protein) gene was used as a target for assay related information about this gene has to be mentioned in the introduction. additionally in the discussion mention why this particular gene has been chosen.

We appreciate the reviewer’s comment, and we added the information about AV1 gene and more in general genome information in the introduction section. Moreover, we discussed why we have chosen the AV1 gene as target for developed assay in the discussion section, as suggested.

  1. For real-time LAMP assay the authors have used two replications rather than three. It will be good to add the results of another replication to enhance the robustness of the study.

We appreciate the reviewer’s comment. For this reason, we performed another replicate improving the results, as suggested.

  1. Lines from 363-366 “specifically……respectively”. Rephrasing is required for clarity better to write the fluorescence rise and plateau of commercial and membrane spots separately for more clarity.

We have rephrased the sentence, as suggested.

Reviewer 2 Report

The reviewed manuscript is dedicated to the design and validation of LAMP-based assay detecting tomato leaf curl New Delhi virus (ToLCNDV), a pathogen causing diseases of horticultural plants. Here, authors designed a real-time LAMP assay for detection of the virus and compared it with the previously devised qPCR techniques. Authors also used the novel method for analysis of plant samples in on-field settings. The presented results are timely and interesting for scientists, specializing on the field of molecular diagnostics. However, several comments need to be made and addressed.

Major issues:

1.      In the Introduction section, authors are encouraged to describe the limitations of currently using methods for detection of ToLCNDV, as similar LAMP-based methods have been reported previously.

2.      Page 4, lines 4-155: “A specific 450-bp nucleotide inside the AV1 gene was chosen as the amplification target.” — was the sequence tested on the presence of nucleotide variations? The appearance of mutations under LAMP primers would reduce the analysis sensitivity. Therefore, it is important to choose the most possible conservative region for primer’s design.

3.      Page 4, lines 185-186: “The real-time LAMP assay was conducted at 65 °C (according to manufacturer’s instructions)” — an experiment with a gradient of elongation temperature would help to increase the efficacy of LAMP leading to better analytical characteristics.

4.      Rapid Sample Preparation Method Suitable for the ToLCNDV real-time LAMP Assay — an analysis of sensitivity is necessary to determine the limitations of the developed extraction method. Plausibly, DNA could be purified from plant tissues spiked with control PCR amplicons, and purified samples could be analyzed by both LAMP and PCR.

5.      Page 13, lines 394-395: “The real-time LAMP assay showed an increase in detecting ToLCNDV-positive samples” — in many cases, LAMP shows a tendency for false-positive results. In that sense, direct sequencing of LAMP amplicons obtained from PCR-negative samples is needed to confirm the ToLCNDV-positive status.

6.      Authors are encouraged to provide a detailed comparison of the devised tests with other similar LAMP-based methods that were reported previously.

Minor issues:

1.      Minor language corrections are needed.

2.      Page 3, line 124: “Tween-20 in one µL of distilled water, pH 7.4” — plausibly, authors described the buffer composition for 1 L.

3.      Page 3, line 132-149. If the description of PCR for detection of ToLCNDV was given in a separate paragraph, it would increase the readability of the manuscript,

4.      Page 4, lines 179-180: “0.2 mM each of ToLCNDV-F3 and ToLCNDV-B3, 1.6 mM each of ToLCNDV-FIP and ToLCNDV-BIP” — plausibly, μM concentrations were used instead of mM.

5.      Table 4 — presenting of DNA concentration in copies per reaction would increase the readability of the manuscript.

6.      Page 9, line 342: “to 4.41 × 105 gnome copies”

7.      Table 6 could be changed to a 2x2 template with rows designating LAMP results and columns representing PCR results. Perhaps, this from could facilitate the understanding of the presented data.

Author Response

Major issues:

  1. In the Introduction section, authors are encouraged to describe the limitations of currently using methods for detection of ToLCNDV, as similar LAMP-based methods have been reported previously.

From lines 432 to 443, we briefly described the limitations of the various techniques in the discussion section compared to the LAMP we developed. Moreover, as suggested by the reviewer, we have added in the introduction section the main limitations of the LAMP techniques already developed for the detection of ToLCNDV (Lines 113 to 122).

  1. Page 4, lines 4-155: “A specific 450-bp nucleotide inside the AV1 gene was chosen as the amplification target.” — was the sequence tested on the presence of nucleotide variations? The appearance of mutations under LAMP primers would reduce the analysis sensitivity. Therefore, it is important to choose the most possible conservative region for primer’s design.

We appreciate the reviewer’s comment. We carried out multiple alignments of ToLCNDV full genome in order to identify the most conserved region, where no mutations are present that could reduce the LAMP specificity and sensitivity. For this reason, in discussion section, we specify that it was selected the ToLCNDV-AV1 coding region as target for primer LAMP design for the absence of recombination events and a very low differentiation in several ToLCNDV isolates on the AV1 gene.

  1. Page 4, lines 185-186: “The real-time LAMP assay was conducted at 65 °C (according to manufacturer’s instructions)” — an experiment with a gradient of elongation temperature would help to increase the efficacy of LAMP leading to better analytical characteristics.

We understand the reviewer’s concern. During the development of the LAMP assay, we carried out different experiments with a gradient in order to increase the efficacy, but we did not get relevant results. For this reason, we preferred to conduct the assay at 65 °C, as optimal temperature for Bst polymerase, recommended by the manufacturer, that allows to reach the best reaction efficacy. Moreover, the specific primer set for the AV1 gene was set up with an assay temperature of 65 °C, using the PrimerExplorer software.

  1. Rapid Sample Preparation Method Suitable for the ToLCNDV real-time LAMP Assay — an analysis of sensitivity is necessary to determine the limitations of the developed extraction method. Plausibly, DNA could be purified from plant tissues spiked with control PCR amplicons, and purified samples could be analyzed by both LAMP and PCR.

We appreciate the reviewer's comment. The sensitivity of the LAMP method has already been addressed starting from the same samples and using the two extraction methodologies but, for the rapid extraction method, we do not understand what the reviewer proposes. In the section "Rapid Sample Preparation Method Suitable for the ToLCNDV real-time LAMP Assay," we simply evaluated the developed technique under field conditions, comparing the results with the conventional PCR technique.

  1. Page 13, lines 394-395: “The real-time LAMP assay showed an increase in detecting ToLCNDV-positive samples” — in many cases, LAMP shows a tendency for false-positive results. In that sense, direct sequencing of LAMP amplicons obtained from PCR-negative samples is needed to confirm the ToLCNDV-positive status.

As the reviewer knows, the LAMP reaction product is composed of fragments of different sizes. Therefore, direct sequencing is particularly complicated. In the tests that were carried out directly in the field and subsequently confirmed under laboratory conditions, the additional steps for the melting curve were performed on all the analyzed samples, in order to find out whether there was a nonspecific reaction that could lead to a false positive result. In this regard, all samples tested positive for ToLCNDV, including those that were asymptomatic and tested positive only by LAMP, displayed the same peak temperature of the melting curve to 84.5 °C, concordant with the ToLCNDV positive controls used in each assay. In order to make the experiment clearer and confirm the positivity of the sample, we added the steps of the melting curve in results sections.

  1. Authors are encouraged to provide a detailed comparison of the devised tests with other similar LAMP-based methods that were reported previously.

We understand the reviewer’s request, but the aim of our work was not to develop a better LAMP technique than the others already developed, but rather a simpler, faster and alternative technique. As reported in the answer to the first question, we described the limitations of the various techniques in the discussion section compared to the LAMP we developed and, in the introduction section, the main limitations of the LAMP techniques already developed for the detection of ToLCNDV without, however, make any kind of comparison with our real-time LAMP assay.

Minor issues:

  1. Minor language corrections are needed.

We revised the entire manuscript in order to improve the English language.

  1. Page 3, line 124: “Tween-20 in one µL of distilled water, pH 7.4” — plausibly, authors described the buffer composition for 1 L.

We have modified the sentence.

  1. Page 3, line 132-149. If the description of PCR for detection of ToLCNDV was given in a separate paragraph, it would increase the readability of the manuscript,

We appreciate the reviewer’s comment and we have adjusted the PCR description in a separate paragraph, as suggested.

  1. Page 4, lines 179-180: “0.2 mM each of ToLCNDV-F3 and ToLCNDV-B3, 1.6 mM each of ToLCNDV-FIP and ToLCNDV-BIP” — plausibly, μM concentrations were used instead of mM.

The concentration of each primer used for the assay optimization is correct.

  1. Table 4 — presenting of DNA concentration in copies per reaction would increase the readability of the manuscript.

We changed in Table 4 the DNA concentration in copies per reaction, as suggested.

  1. Page 9, line 342: “to 4.41 × 105 gnome copies”

We have corrected the phrase “to 4.41 × 105 genome copies”

  1. Table 6 could be changed to a 2x2 template with rows designating LAMP results and columns representing PCR results. Perhaps, this from could facilitate the understanding of the presented data.

We appreciate the reviewer’s advice, but we believe that the table should be presented in this form, in order to make the reported results more exhaustive.

Round 2

Reviewer 2 Report

Many thanks to authors for their detailed reply on the comments and changes of the manuscript. However, several comments still need to be made and addressed.

Major issues:

1.      “We appreciate the reviewer's comment. The sensitivity of the LAMP method has already been addressed starting from the same samples and using the two extraction methodologies but, for the rapid extraction method, we do not understand what the reviewer proposes. In the section "Rapid Sample Preparation Method Suitable for the ToLCNDV real-time LAMP Assay," we simply evaluated the developed technique under field conditions, comparing the results with the conventional PCR technique.” — spiked controls meant plant samples with a various amount of ToLCNDV PCR-amplicons. Thus, knowing the amount of added amplicon, it is possible to determine the efficacy of purification procedures. It would underline the robustness of the developed rapid method.

Minor issues:

1.      “The concentration of each primer used for the assay optimization is correct.” With all respect to authors, it seems that primers in LAMP are used in µM concentrations.

As https://www.neb.com/protocols/2014/06/17/loop-mediated-isothermal-amplification-lamp

Author Response

Major issues:

  1. “We appreciate the reviewer's comment. The sensitivity of the LAMP method has already been addressed starting from the same samples and using the two extraction methodologies but, for the rapid extraction method, we do not understand what the reviewer proposes. In the section "Rapid Sample Preparation Method Suitable for the ToLCNDV real-time LAMP Assay," we simply evaluated the developed technique under field conditions, comparing the results with the conventional PCR technique.” — spiked controls meant plant samples with a various amount of ToLCNDV PCR-amplicons. Thus, knowing the amount of added amplicon, it is possible to determine the efficacy of purification procedures. It would underline the robustness of the developed rapid method.

We understood the reviewer’s advice. The test suggested by the reviewer was performed, and the results obtained were comparable with those obtained by the DNA extraction method using commercial kits. In paragraph 2.7 and Table 6, we included the analyses of laboratory conditions, using both total DNA and membrane spot crude extract, and we have obtained comparable results, with a minimal delay in the increase of fluorescence and reaction plateau. Nevertheless, we prefer not to include the additional test suggested by the reviewer, because we did not find any scientific papers in the literature, where the sensitivity test is performed using the rapid extraction method, as the sensitivity of the developed assay has been evaluated previously in paragraph 2.5, through quantification of the minimum number of molecules detectable by the assay. In our opinion, by performing the test suggested by the reviewer, the in-field conditions cannot be simulated. In addition, tests with field samples demonstrated the robustness of the rapid extraction method under both in-field and laboratory conditions, using both DNA extraction methods.

Minor issues:

  1. “The concentration of each primer used for the assay optimization is correct.” With all respect to authors, it seems that primers in LAMP are used in µM concentrations.

As https://www.neb.com/protocols/2014/06/17/loop-mediated-isothermal-amplification-lamp

We apologize to the reviewer and realized that we made an error during the writing process. The sentence has been changed to the correct concentrations, as suggested.